



# First 2D record of a tsunami by SWOT satellite: observation data and preliminary numerical simulation of the 19 May 2023 tsunami near the Loyalty Islands

Jean H. M. Roger[1], Yannice Faugère[2], Hélène Hébert[3], Antoine Delepoulle[4] and Gérald Dibarboure[2]

[1]GNS Science, Earth Structure and Processes, Lower Hutt 5011, Aotearoa New Zealand
[2]Centre National d'Etudes Spatiale (CNES), Toulouse 31400, France
[3]CEA, DAM, DIF, Arpajon 91297, France
[4]Collecte Localisation Satellites (CLS), Ramonville-Saint-Agne 31520, France

*Correspondence to*: Jean H.M. Roger (j.roger@gns.cri.nz)

**Abstract.** The NASA-CNES altimetry mission SWOT (Surface Water and Ocean Topography) deployed in December 2022 embarks a Ka-band Radar Interferometer (KaRIn), providing a 120 km-wide swath sea level measurement. On 19 May 2023, SWOT was able to record a 2D signature of the tsunami generated by the Mw 7.7 earthquake southeast of the Loyalty Islands (southwest Pacific Ocean), about 1 hour after the earthquake, on a straight SSW-NNE path. Comparison between numerical models and real measurements was performed to assess SWOT's ability to monitor tsunami waves. A uniform 15 coseismic slip rupture model allows to satisfactorily fit the regional observations. Testing models against a dynamic representation of the tsunami wavefield (instead of static) show a good phase agreement, but simulated amplitudes and energy spectra are lower than the measurements. However, this SWOT unprecedented 2D observation critically inform on tsunami propagation and modeling, and offer a breakthrough perspective for better predictions.

## 1 Introduction

The probability of detecting a tsunami with a satellite altimeter is low, since the satellite must orbit, by chance and within a few minutes, over the propagating tsunami wavefield, taking into account tsunami propagation speed and direction, and waves dispersion. However, several cases have been reported in the literature: the first observation of a tsunami propagating at the surface of the oceans using satellite altimetry was performed by TOPEX which measured the tsunami triggered by the 1992 Mw 7.6 earthquake in Nicaragua (Okal et al., 1999). But it is only with the 2004 Indian Ocean tsunami that satellite 25 altimetry data provided useful additional information about tsunami propagation and amplitudes (Ablain, 2006, Hayashi, 2008, Wang and Liu, 2006, Gower 2007, Godin et al. 2009). In that specific case the resulting large tsunami amplitude induced clear signal visible on both satellite and tide gauge records throughout the Indian Ocean. It was the first tsunami to be detected very clearly in the sea level anomaly derived from altimeter measurements from 4 satellites crossing its wavefront (TOPEX, Jason-1, Envisat and GFO; Gower, 2007). With such a rich constellation, the separation of the tsunami 30 signal from the oceanic variability was possible. The maximum wave amplitude observed in the deep ocean was about 60 cm



on 3 different satellite tracks 2 hours after the earthquake occurrence. This unprecedented record allowed the recovery of a seismic rupture slip model based on a single inversion of offshore tsunami data, which was very consistent with models obtained from seismic and geodetic data (Sladen and Hébert, 2008). Other signals at small wavelength (20-40 km) but strong amplitude (20-25 cm) were discovered in the full rate altimetry dataset (600 m posting for TOPEX and 300 m for Jason and

Envisat). These signals, in agreement with the theory of tsunami wave propagation, were not seen in the numerical model outputs, demonstrating again the high value of altimetry information. More recently, altimetry analyses were also performed in the case of the 2010 Chile (Hamlington et al., 2011) and 2011 Tohoku (Hamlington et al., 2012) tsunamis. Finally, in January 2022, the tsunami generated by the eruption of the Hunga Tonga–Hunga Ha'apai submarine volcano in Tonga (southern Pacific Ocean) was measured by various in situ, airborne and space platforms (e.g., Shrivastava et al., 2023)

including the Chinese altimeter satellite HY2B which recorded, 1 hour after the eruption, a 20 cm amplitude tsunami signal on a track between the Kermadec and Samoa Islands (Faugere et al., 2022). The oceanic variability processing was necessary to refine the interpretation.

On 16 December 2022 a new satellite, named SWOT for Surface Water Ocean Topography, was deployed as a result of a collaboration between NASA and CNES, the US and French Space Agencies, respectively, in cooperation with the United

Kingdom and Canadian Space Agencies (UKSA and CSA, respectively; Vaze et al., 2018). The SWOT satellite has completed the international altimetry constellation, featuring an innovative feature compared with its predecessors: the wide swath. The mission's objective is to make the first global survey of Earth's water surfaces: ocean surface topography, inland water heights, river discharge, etc. (Morrow et al., 2019; Fu and Rodriguez, 2004). To achieve this goal, the main instrument of the SWOT mission is a Ka-band interferometer (KaRIn) that delivers a bi-dimensional (2D) view of the water surface

topography. To meet its requirement, the interferometer requires an extremely precise attitude and orbit control system (AOCS), as well as the full suite of "conventional" altimetry payload: a nadir altimeter (POSEIDON-3C, TOPEX/Jason class), precise orbit determination sensors (DORIS positioning system, precise Global Navigation Satellite System, laser reflector array) and a microwave radiometer (to correct for the wet troposphere path delay). The two important SWOT features over the ocean are (1) its ability to provide a synoptic 2D view of the ocean surface without interpolation, and (2) its

high precision (vertical accuracy of several cm and up to 1 km of spatial resolution over oceanic surfaces; Archer et al., 2025). This combination makes it possible to capture many ocean features such as mesoscale or internal waves, as small as a few kilometers, including the wavelength range of tsunami waves. Fu et al. (2024) illustrate the breakthrough provided by SWOT during its first months of operations.

Today the SWOT satellite provides global coverage up to a latitude of ±78° with an exact repeat orbit of 21 days. However,

SWOT was first operated on a 1-day repeat orbit from April up to July 2023, for calibration and validation purposes. Validation phase ended in spring 2024 with the conclusion that SWOT is meeting its requirements (Dibarboure et al., 2025). Peral et al. (2024) provide an in-depth review of the good behavior of the KaRIn instrument and preliminary analyses report a nominal behavior of the product quality (Raynal et al., 2023, Bohé et al., 2023, Chen et al., 2023, Fjörtoft et al., 2023). During this phase, it provided not only the first 2D water surface topography of all water surfaces, but also the first daily



revisit of topography by any satellite altimeter mission. The high temporal revisit implies very sparse spatial sampling, which makes it all the more incredible how fortunate it was to capture the specific tsunami signature presented herein.

On 19 May 2023 at 02:57:03 UTC (Te hereafter), a magnitude Mw 7.7 earthquake occurred in the south of the Vanuatu Subduction Zone (VSZ), ~100 km southwest of Matthew Island and ~320 km southeast of Maré Island, Loyalty Archipelago (Figure 1). The different published focal mechanism solutions show a strongly non-double couple normal fault rupture,

located on the plunging Australian plate, on the outer-rise of the VSZ (Roger et al., 2025). The earthquake epicenter showing a relatively shallow depth (~18 km; https://earthquake.usgs.gov/earthquakes/eventpage/us6000kd0n/executive), the seismic rupture implying a significant vertical motion of the sea bottom triggered a tsunami recorded first on the tide gauge located in Tadine Harbour, Maré Island. It was then progressively recorded on many other coastal and deep-ocean sensors of the southwest Pacific Region, including the DART systems (Deep-ocean Assessment and Reporting of Tsunamis; e.g., Meinig et

al., 2001) of the recently deployed New Zealand network (Power et al., 2018), as far as New Zealand, Tonga or Tasmania (Robert et al., 2024; Roger et al, 2025).

The SWOT satellite flew over the region around 4:00 UTC, approximately 1 hour after the earthquake occurred, capturing the tsunami signature along a SSW-NNE pathway between 30°s and 17°S of latitude. This synchronized overflight was highly improbable, given the infrequency, propagation speed, and dispersion of these events, especially in the deep ocean, as

it is the case for all of the tsunamis recorded via altimetric missions. However, since the SWOT satellite deployment, at least one seiche (Monahan et al., 2025) and three tsunamis have been recorded, including the Loyalty Islands tsunami, the tsunami triggered by the 2 May 2025 Mw 7.4 earthquake having occurred within the Drake Passage (south of Patagonia, Chile: https://www.aviso.altimetry.fr/en/missions/current-missions/swot/portfolio-of-swot-first-results/tsunami-waves-observed-by-swot-for-the-second-time.html) and the tsunami triggered by the 30 July 2025 Mw 8.8 earthquake offshore Kamchatka

Peninsula (Russia: https://www.aviso.altimetry.fr/en/missions/current-missions/swot/portfolio-of-swot-first-results/a-third-tsunami-observed-by-swot-due-to-kamchatka-earthquake.html). We will only focus on the 2023 Loyalty Islands tsunami in the rest of this article.

## 2 Earthquake rupture model and simulation of tsunami

Numerical simulation of tsunami has been performed using COMCOT (Cornell Multi-grid Coupled Tsunami model), a

robust modelling code tested and widely applied to numerous tsunami studies for about four decades (e.g. Liu et al., 1995; Wang & Power, 2011; Wang et al., 2020; Roger et al., 2023; Roger and Wang, 2023). It computes tsunami generation, propagation and coastal interaction by solving both linear and nonlinear shallow water equations using a modified explicit leap-frog finite difference scheme and considering the weak dispersion effect (Wang, 2008). The initial sea surface deformation is calculated using the Okada (1985)'s formulae with the fault plane geometry and either a uniform or non-

uniform slip distribution. Water surface elevation and horizontal velocities are calculated respectively at the cell center and



at the edge centers of each grid cell of the computational domain. Absorbing boundary schemes are used at the boundaries of the computational domain to dampen the incoming waves, avoiding reflection from the grid boundaries.

For the purpose of the present study, numerical simulation of tsunami generation and propagation have been computed over a 15 arcsec-resolution grid covering the southwest Pacific Region using GEBCO (2023) 15 arcsec-resolution dataset. A
simulated propagation time of 3 hours includes the passage of the satellite over the region. This resolution is considered sufficient at first order to simulate a tsunami in the deep ocean and compare with SWOT records.

**Figure 1: The 19 May 2023 Mw 7.7 earthquake and tsunami. Tsunami travel times from the epicenter were calculated using**
**Mirone software (Luis, 2017). The coloured band symbolizes the swath track of the SWOT satellite dataset presented herein with T0 and T1 symbolizing the start and end time of the swath. VT: Vanuatu Trench; LR: Loyalty Ridge; NR: Norfolk Ridge. Background topography is from GEBCO (GEBCO, 2023).**



The selected source for the tsunami generation is coming from the analysis done by Roger et al. (2025). It is a GCMT-based scenario (Global Centroid Moment Tensor project; Dziewonski et al., 1981, Ekström et al., 2012) showing the best fit in

terms of wave phase and amplitude and arrival time between the simulated and real recorded waveforms at DART systems and coastal gauges location. The parameters of this scenario are based on the GCMT focal mechanism solution with hypocentral longitude, latitude and depth being 170.89°, -23.35° and 20.4 km, respectively. The selected nodal plane is the one showing strike = 274°, dip = 47° and rake = -71°. Empirical relationships from Strasser et al. (2010) and Wells and Coppersmith (1994) were used to calculate a 95 km-length and 48 km-width fault plane, as well as a 1.7 m-coseismic slip.

## 115    3 Extraction of the tsunami signal from SWOT measurements

On 18 May 2023, SWOT flew over the area disturbed by the tsunami waves from the southwest to the northeast exactly between 3:59 UTC and 4:02 UTC. This corresponds to pass number 147 of cycle number 525 of the SWOT satellite. SWOT Products derived from both SWOT KaRIn and Nadir measurements from 19 May 2023 (pass 147 of cycle 524 -before the tsunami- and 525 -during the tsunami-) have been used. Unfortunately, the day after no measurement is available due to a

platform anomaly. The area captured during this time is shown on Figure 1.

Additionally, 7 Nadir altimeters satellites, namely Sentinel 6 MF, Jason-3, Sentinel3a, Sentinel3b, Altika, Cryosat-2 and HY2B are used to provide ocean large scale and mesoscale data, before and after 19 May, to compute the Oceanic variability content in order to extract the tsunami signal from SWOT. The Nadir products used in this study are Level-3 (L3) products, provided by the multi-mission Data Unification and Altimeter Combination System (DUACS), the multi-satellite, multi-

agency system developed by CNES and CLS operated since 2015 within the Copernicus Marine Service (https://data.marine.copernicus.eu/product/WAVE_GLO_PHY_SWH_L3_MY_014_005/description). The standards of these L3 products, available through the Copernicus Marine Service catalogue, refer as DUACS DT 202 described in Taburet et al. (2019).

The DUACS system has been upgraded by CNES in the recent years 2016-2023 in order to be able to handle SWOT

measurements over the ocean for various scientific and operational applications. L3 algorithms and products are developed specifically for SWOT/KaRIn (geophysical corrections, t, cross-calibration, editing, filtering steps). In particular, the calibration used at L3 (Ubelman et al., 2023, 2024) ensures a good homogeneity between SWOT and the Copernicus Marine Service Nadir product. SWOT "2 km" L3 products version 1.0.2 available on Aviso (https://www.aviso.altimetry.fr/en/data/products/sea-surface-height-products/global/swot-l3-ocean-products.html)    are

derived and used in this study (see Dibarboure et al. [2024] for further details).

To extract the finest estimation of tsunami signal from altimeter measurements, a specific processing is necessary. First a Sea Level Anomaly (SLA) is obtained after instrumental and geophysical correction (e.g., tide signal, or ocean response to atmospheric pressure and wind). The L3 variable "ssha" containing the SWOT 2D SLA includes many different ocean signals such as large scale and mesoscale ocean variability which have to be removed. To this end, the MIOST (Multivariate



Inversion of Ocean Surface Topography) mapping technique (Ubelmann et al., 2021, 2022) was used to map these oceanic

signals not related to the tsunami, interpolating data in space and time along the 19 May 2023 SWOT profile. These

interpolated SLA data correspond to the sea level signals that would have been observed on 18 May, when the tsunami had

not occurred yet. This algorithm is applied to the 7 Nadir satellite altimeters in order to produce the best signal estimation.

Correcting SWOT SLA from the MIOST estimation allows to analyze the tsunami signal with a good quality, thanks to a

fine space/time sampling of the ocean with 7 altimeters. However, the correction is not perfect at fine scales, as detailed in

Ballarotta (2019 and 2023). The effective spatial resolution of MIOST mapping is around 200 km in the Loyalty Islands

area, meaning that oceanic features with wavelength below 200 km, may still be present in SWOT measurement, as other

non-tsunami signals, notably near the coasts, due to imperfect tidal corrections or atmospheric effects.

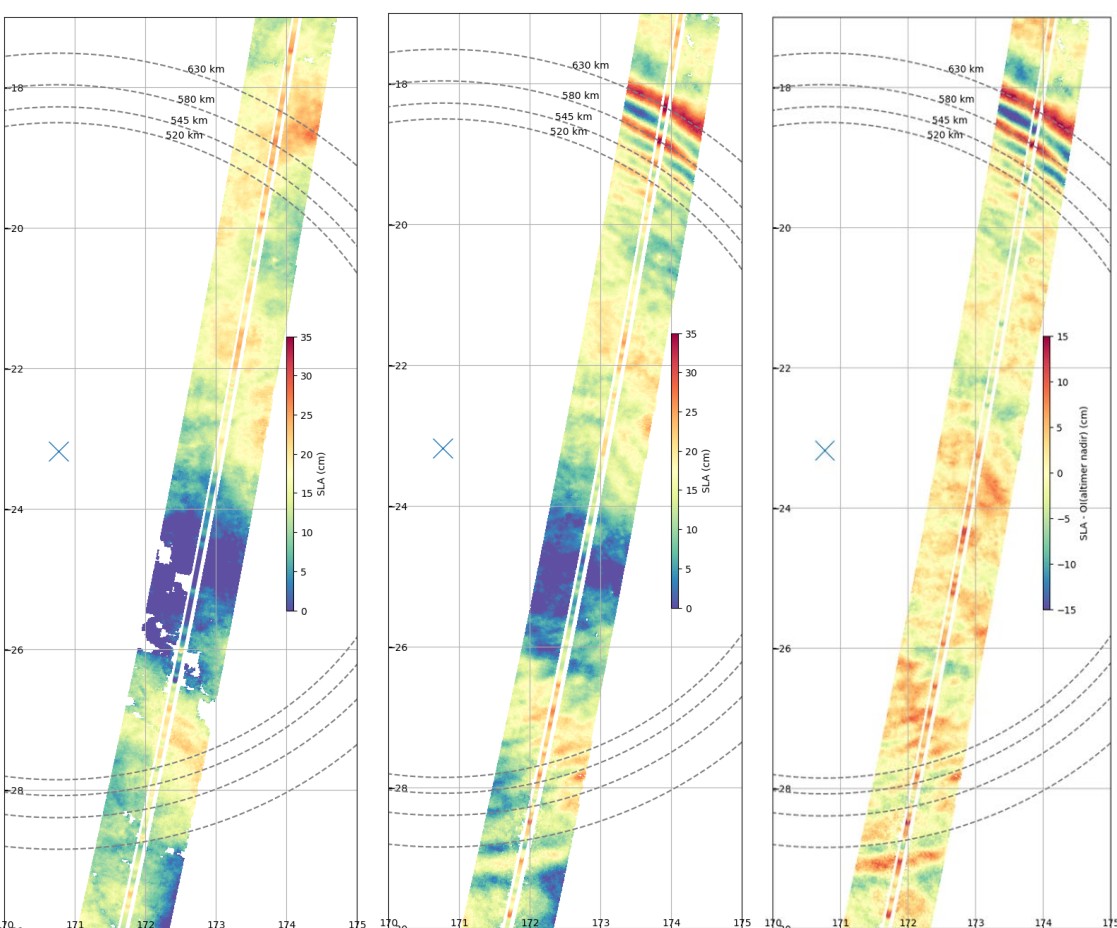

**Figure 2: The SWOT SLA on pass 147 (left) the 18th of May 2023 (cycle 524) 23h hour before the Earthquake, (center) the 19th of May (cycle 525) 1h after the earthquake. (right) is the same as (center) but with ocean variability estimated by MIOST removed. The dashed concentric lines symbolize the distance from the epicenter.**

Figure 2 shows the SWOT SLA on pass 147 (Figure 2a) first on the 18th of May 2023 (cycle 524), ~23 hours before the

earthquake, and (Figure 2b) secondly the 19th of May (cycle 525) ~1 hour after the earthquake. The two figures show similar



patterns, notably large oceanic and mesoscale signals. The two main wavefronts, called later the North and South wavefront of the tsunami signature, are visible respectively on (b) at the latitudes 18°S and 29°S. First, a good consistency between swath and Nadir SLA (central line) is observed. On Figure 2c the MIOST estimation is removed, better revealing the tsunami signature. The North wavefront measured at 630 km North away from the earthquake epicenter appears clearly. The South wavefront is lower in amplitude. Note that the two wavefronts have not been measured at the same time. The satellite heading North met the South Front at 3:59 UTC and the North front ~3 minutes later at 4:02 UTC (T0 = Te +62' and T1 = Te + 65'on Figure 1, respectively). Secondary waves within the 630 km-radius circle around the earthquake epicenter are visible. Contrarily to a simple 1D field the 2D measurement allows to easily discriminate the waves from the residual oceanic variability.

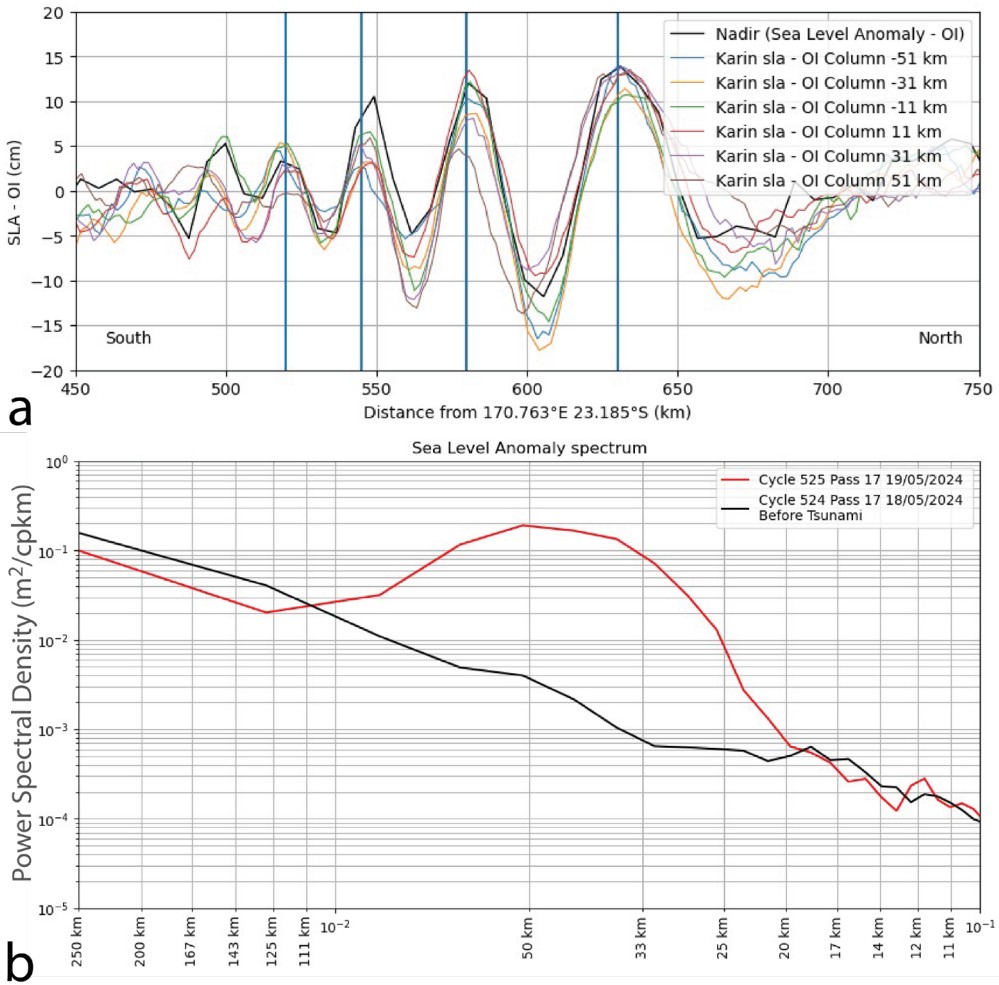

**Figure 3: SWOT SLA and Power Spectral Density. (a) Zoom on SWOT SLA pass 147 on 19 May in 1D along track of the Nadir and Karin instruments for several across track distance, at 51, 31 and 11km either side of the nadir position. (b) Power Spectral Density of SWOT SLA over the same area for 19 and 18 May (the optimal interpolation OI is the MIOST).**



Figure 3a focuses on the north front wave showing 1D along-direction profiles of both the nadir and swath SLA (KaRIn instrument). Several profiles are displayed, for different track distances, at +/- 51, 31 and 11km, on either side of the nadir

position. The first front wave on figure 3b shows the largest amplitude up to 15 cm, and the largest wavelength of more than 100 km. Moving closer to the epicenter, thus later in the propagation pattern (toward the left of the figure), the amplitude and the wavelength decrease to a few cm and a few tens of km, respectively. This is expected as the short wavelengths travel slower than the longer wavelengths due to tsunami frequency dispersion. Figure 3c shows the spectral analysis of SWOT SLA over the same area for the 18th and the 19th of May. A large energy lobe is clearly visible at wavelengths between 20

and 120 km documenting the strong dispersive property of the tsunami signal.

## 4 Comparison between measurement and model and discussion

### 4.1 Comparison

Figure 4 shows (a) the simulated tsunami propagation wavefield 1 hour after the earthquake occurrence time and (b) a superposition of the satellite swath over this simulated wavefield, focusing on the northern region of the concerned area.

There is a good correlation between the two profiles regarding the timing location (however showing a slight delay of the simulated main peak compared to the SWOT record) of the tsunami wavefront and the phase of the three main first waves. However, the differences between the observation and the model are notable in terms of amplitude, approximately with a factor of 2, while a better fit could be obtained, such as in a previous work on the 2004 Indian Ocean tsunami (Ablain, 2006; Sladen and Hébert, 2008). In this latter case the earthquake source was however more refined. In addition, the dispersion of

the tsunami waves is very visible on 2023 SWOT data (Figures 3a, 4b), and although our model accounts for dispersion, it remains preliminary, and does not produce enough amplitudes for the dispersed wavetrain.

The tsunami simulation results are compared with the swath of the satellite on the whole area using shade colours on Figure 5. Figure 5a shows a superposition of the satellite swath over the simulated tsunami propagation wavefield 62' after the earthquake occurrence time Te. For this exercise, the swath dataset was resampled to a regular 15 arcsec-resolution grid

using kriging interpolation to fit the simulation grid spatial resolution. A first impression shows a really good correlation between the two datasets, especially for what concerns the location of the tsunami wavefronts on the southwest and northwest of the region. Generally, this also allows to highlight some finer consistency notably concerning secondary waves which show also a good agreement, with sometimes some marginal differences, considering the limited knowledge and possible complexity of the source (the signal is not reproduced perfectly on all DART systems and coastal gauges as

indicated by Roger & Gusman, 2024), the resolution of the bathymetry (15 arcsec, with some variations of quality depending on the region), and the general challenging processing of the SWOT dataset (applied corrections, artifacts, etc.).





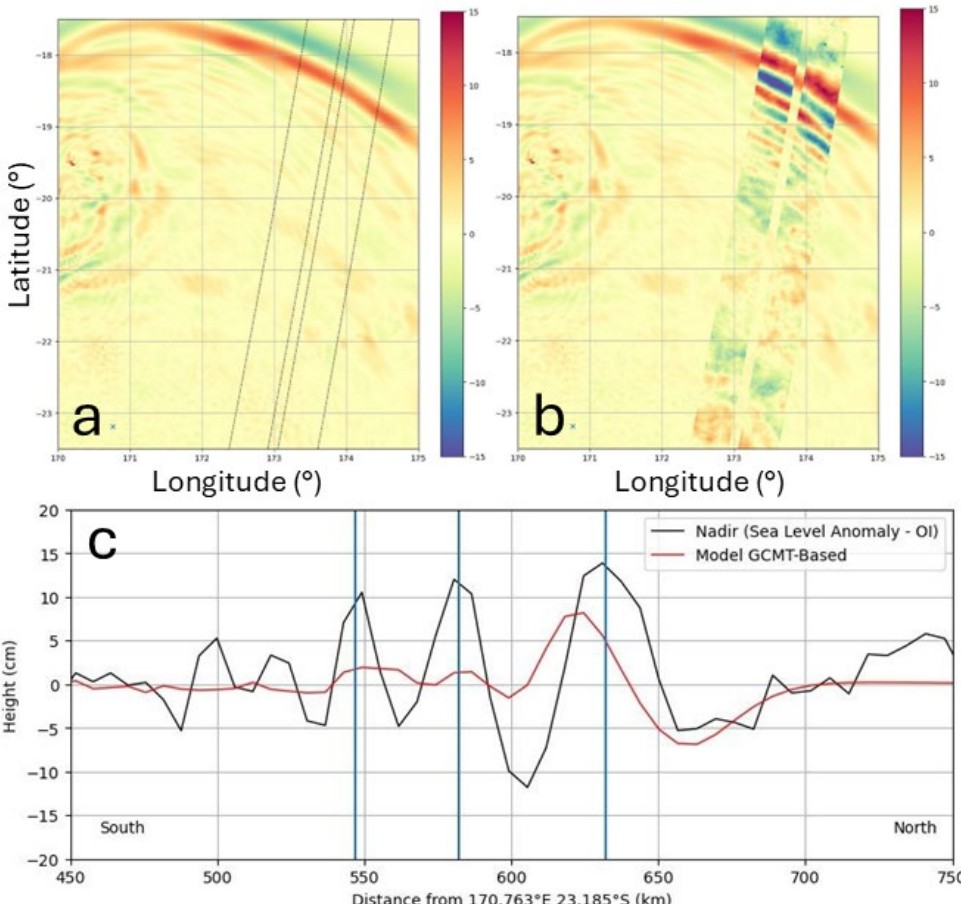

**Figure 4: Comparison between nadir SWOT measurements and the model estimation cross-section at T+1h: (a) GCMT-based model only with position of SWOT measurement, (b) SWOT measurement superimposed on the model field, (c) SWOT and model comparison along the nadir.**

Figure 5b shows the difference between the satellite record and the simulation result specifically on the swath band within a difference range of -8 cm to 39 cm, with a median value of 12 cm. First, it highlights that the satellite data is larger than the simulated signal on most of the swath region. Then, it shows that the larger difference sits clearly in the northeastern wave front. Note that it takes less than 5 minutes for the satellite to cover the region where the tsunami was propagating, recording both the southwestern and northeastern fronts of the tsunami. The median value of the ocean depth along the nadir track is ~3000 m, which leads to a tsunami speed of approximately 620 km/h. In five minutes, the tsunami would have travelled ~50km. The simulation results shown on figure 5 corresponds to an instantaneous snapshot of the sea surface at t=1h, which is an approximation of what the satellite was able to measure during its 5-min journey above the region happening between 0h57min51.6s and 1h2min30.9s after the earthquake time. Thus, the simulation output is too late in the southwest and too early in the northeast, which can explain partially the differences.



**Figure 5: (a) Superposition of the SWOT satellite swath (presented on figure 2) with the simulated tsunami propagation wavefield at t=T0 (62' after earthquake occurrence time Te), using the GCMT-based scenario of coseismic rupture from Roger and Gusman (2024); (b) sea-surface elevation difference between SWOT record and the modelling output. The red line symbolizes the Nadir location in the middle of the swath (KaRIn). Note that SWOT record goes from T0 = Te + 62' (SW) to T1 = Te + 65' (NE).**

## 4.2 Comments on the misfit between tsunami observation and model

Source models are considered here as badly constrained, due to the lack of seismic stations and additional sensors like GNSS stations leading to bad azimuthal coverage of earthquakes occurring there, and also due to the limited knowledge of the tectonic processes in this specific region of the VSZ. Therefore, the signal is generally not reproduced perfectly on all DART systems and coastal gauges available (Roger et al., 2025). This led Robert et al. (2025) to move the source epicenter ~20 km from its calculated region to fit the records a bit better. Thus, the complexity of the 19 May 2023 event, which was largely non-double couple and located in a complex tectonic region, leads to real difficulties to correctly reproduce the associated tsunami generation mechanism.

For deep ocean tsunami modeling, the resolution of the bathymetry used here (15 arcsec, equivalent to ~450 m), is sufficient to account for tsunami long waves (50-100 km), even for presumably dispersed waves as shown on Figure 4 (~10 km long).



However as underlined by Robert et al. (2025), the highly complex bathymetry of this region may also slightly impact the results at small scale (secondary waves reflection, diffraction, etc.).

### 4.3 Sources of the differences from SWOT data analysis

Despite the novelty of the instrument onboard SWOT, i.e. KaRIn, the results from the validation phase indicate that its
measurement can be trusted, reinforced by a consistent SLA obtained on the classic Nadir instrument also onboard SWOT. However, SWOT signals can contain, as mentioned earlier, residual errors due to imperfect geophysical and oceanic corrections. Although we can definitely conclude on its ability to detect/record tsunami waves along a 2D swath, the validation of the wave amplitudes may need further consideration, including comparison of records with simulation outputs for other future observations.

A Mean Sea Surface (MSS) is also used to correct SWOT measurements from the geoid signature. This MSS model is very recent (Laloue, 2024), meaning that unresolved geoid signatures may remain, polluting SWOT products at short wavelengths. This may happen especially in this area of rugged bathymetry, with uncharted seamounts, rifts, continental shelf, etc.

### 5 Conclusion & perspectives

For the first time on 19 May 2023, tsunami waves in the open ocean have been clearly measured in 2D by the SWOT satellite deployed in 2022. The recorded signal well describes the tsunami, showing its wave amplitudes and wavelengths. 2D data allows to better discriminate the waves and related errors from the ocean signal and properly distinguish fine variations within the swath. Preliminary numerical simulations of tsunami have been computed using COMCOT. There is a good correlation between the two profiles regarding the location of the tsunami wavefront. However, the differences between
the two signals are notable in terms of amplitude, by approximately a factor of 2. This highlights that (1) the source model is not well constrained, notably with this kind of complex earthquake source case with a significant non double couple component; (2) the azimuthal gap of measurement needs to be filled in for this seismically active section of a complex subduction zone; and (3) future surveys may focus on completing the bathymetry gaps, especially in the North Fiji Basin.

This study demonstrates the opportunity of using SWOT to contribute to a better understanding and improvement of the
modeling of tsunami generation, propagation and dissipation. Observations can be used, in particular, to refine the initial displacement conditions due to the earthquake, and, during the propagation phase, provide an unprecedented picture of the dispersed tsunami waveforms. The recent May 2025 Drake Passage and July 2025 Kamchatka tsunamis have underlined the usefulness of such records to better refine tsunami sources and fit the modeling results to current records. Interest in such measurements can also be highlighted, to better confirm tsunami triggered by an earthquake or another geohazard in
complement to DARTs, coastal gauges, seafloor cables and GNSS-equipped vessels. it reminds the importance of using multisensors array to better understand and model complex tsunamis. This kind of approach also allows in turn to better



understand oceanic variability measured on these modern altimetry satellites, and to better correct its signature to study oceanic hazards. In any case, due to the kinematic conditions of satellite path and tsunami propagation, this kind of observation remains indeed very rare and probably not well adapted for the tsunami warning challenges (see also Hébert et

al., 2020 for a review).

Whatever the misfit regarding the earthquake source models, it is also crucial to stress that well-designed seismic networks are still essential to contribute to efficient seismic warning in the first minutes after an earthquake. While regional initiatives are already set up or in preparation in situ thanks to several national commitments (for instance, Oceania Regional Seismic NETwork, ORSNET, developed since the 2010s; or the New Zealand regional DART network, fully operational since 2021:

Power et al., 2018), efforts must be maintained in the long term to grant timely access to reliable and complete datasets. In addition, submarine data acquisition is more and more developed and should growingly contribute to early warning, with new projects such as SMART cable (Howe et al., 2024) that are able to monitor both seismic activity and tsunami propagation with dedicated networks of in situ seismic sensors and sea-level gauges.

There is a need to rework the model, especially the seismic rupture accuracy with a better understanding of the event in

general, in order to improve the quality of the amplitude of the front waves. SWOT data may be used to assess the quality or accuracy of the source model, but also provides an unvaluable look at the secondary waves.

In addition, SWOT data could also be integrated in an inversion process as it is already the case for static deep-ocean and coastal gauges datasets. Although inverting data on a dynamic 2D track instead of waveforms from time series at specific locations might be challenging, numerical models performed on discrete sampling points along the track, accounting for the

satellite motion, could be integrated in massive inversion process using advanced numerical techniques (e.g., machine learning).

Finally, the next generation of wide-swath satellites is already in the starting-blocks after SWOT, i.e. the S3NG-T constellation (Copernicus programme, e.g., Le Traon et al., 2025) and should be deployed ~2030. Although repeatability will probably still not be sufficient for real-time early detection and warning, this study shows how such observation improves

the 2D picture of a propagating tsunami and can contribute to enhanced studies of tsunami physics and modeling offshore

**Code availability**

COMCOT code is available upon request from Dr Xiaoming Wang (Tsinghua University, Beijing, China).

**Data availability**

The Level-3 (L3) SWOT products version 1.0.2 used in this study are available on the Aviso repository from CNES

(https://www.aviso.altimetry.fr/en/data/products/sea-surface-height-products/global/swot-l3-ocean-products.html).



**Author contribution**

JHMR: Conceptualization, Formal analysis, Investigation, Validation, Visualization, Writing (original draft preparation)

YF: Conceptualization, Formal analysis, Investigation, Validation, Writing (original draft preparation)

HH: Investigation, Writing (original draft preparation)

AD: Data curation, Formal analysis, Investigation, Visualization

GD: Data curation, Formal analysis, Investigation, Visualization

**Competing interests**

The authors declare that they have no conflict of interest.

**Acknowledgements**

The authors are grateful with Dr Xiaoming Wang for providing the last version of COMCOT tsunami modelling code and unwavering support to use it. The study represents a fruitful scientific collaboration between GNS Science (New Zealand), CNES, CEA and CLS (France).

**Financial support**

This study has been partially funded by GNS Science's Hazard and Risk Management research programme (MBIE, New

Zealand, Strategic Science Investment Fund, Contract C05X1702).

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
