# Peer review of "First 2D record of a tsunami by SWOT satellite: observation data and preliminary numerical simulation of the 19 May 2023 tsunami near the Loyalty Islands"

_EGUsphere, 2025_

## Referee Comment (RC1)

The work entitled "First 2D record of a tsunami by SWOT satellite: observation data and preliminary numerical simulation of the 19 May 2023 tsunami near the Loyalty Islands" focuses on satellite radar observation of a tsunami event on 19 May 2023. It is the first dynamic observation of a two-dimensional tsunami wave field. The satellite's orbit allowed the observation of the wavefront propagating in both directions from the epicenter, about 1 h after the earthquake. I wonder why the authors call it 2D observation, when in fact it is a 3D observation because it includes the sea surface elevations (Sea Level Anomaly) within a twodimensional domain. These observations are compared with numerical simulations performed with COMCOT model, showing generally good agreement with observations, although some differences reveal weak points of the simulations and indicate that there is room for future model improvements. Simulations show a small delay in the main peak of the tsunami wavefront and underestimated amplitudes, especially for the secondary waves in their propagation to the north. The results presented are relevant and highlight the usefulness of SWOT data to improve tsunami predictions. Some paragraphs and figures show inconsistencies that must be corrected (see attached file). The manuscript presents new data, and I recommend publication once the minor changes have been addressed. I hope my comments are useful to the authors and help improve the quality of the paper.

**Specific comments:**

**Title and abstract:** Why 2D record? It is an image of a 2D domain, but it includes the sea level anomaly, which is the third dimension, therefore, perhaps it could be called a 3D record?

Lines 116 to 119: This paragraph starts "On 18 May 2023".... "This corresponds to pass number 147 of cycle 525". The tsunami occurred on 19 May, so I think this date is mistaken. And then, in line 118 it says "measurements from 19 May 2023 (pass 147 of cycle 524 ... and 525 ...)" this part must be clarified, as cycle 524 and 525 should correspond to different dates (are those 18 May and 19 May?).

**Caption of Figure 3:** "(the optimal interpolation OI is the MIOST)" Please include this "optimal interpolation" in figure 3b and explain it if necessary, as it is not clear in its current form.

**Section 4:** Please include an introduction paragraph between lines 176 and 177 indicating the contents of section 4.

**Line 188:** "tsunami propagation wavefield 62' after the earthquake". It indicates that figure 5 shows the simulation 62' after the Te but in line 207 it says that the results are shown "at t=1h". Clarify which time was used.

**Lines 206 and 207:** "In five minutes, the tsunami would have travelled ~50 km". Include a discussion about the lack of phase agreement in figure 4c. If the north wavefront was observed at 1h 2 min, the difference in phase of around 20 km (for the first wavefront) may be easily explained.

**Section 4.1:** In general, I think this section would be better structured if the results of figure 5 were shown first followed by the zoomed-in view of figure 4 on the north wavefront. Additionally, the results and analysis performed in figure 4 could be improved by showing the results of the COMCOT model at time 1h 2min which will likely achieve better agreement with the north wavefront observation.

**Technical corrections:**

**Figure 2 and lines 153 to 157:** The caption of figure 2 refers to the different panels as left, center, and right, while the text in the next paragraph refers to 2a (line 153), 2b (line 154), b (line 156) and 2c (line 157). Please include the letters in figure 2, correct the caption, and line 156 accordingly.

**Lines 170 and 173:** References to figure 3b and 3c. It seems that they should refer to 3a and 3b, respectively. 3c does not exist.

Figure 3: Please include the title of the X-axis in panel b.

**Line 180:** It explains the results shown in figure 4c without introducing the contents of this panel c before. Please include a description of what the profiles of panel c are.